# Unexpected Decarbonylation of Acylethynylpyrroles under the Action of Cyanomethyl Carbanion: A Robust Access to Ethynylpyrroles

**DOI:** 10.3390/molecules28031389

**Published:** 2023-02-01

**Authors:** Denis N. Tomilin, Lyubov N. Sobenina, Alexandra M. Belogolova, Alexander B. Trofimov, Igor A. Ushakov, Boris A. Trofimov

**Affiliations:** 1A.E. Favorsky Irkutsk Institute of Chemistry, Siberian Branch, Russian Academy of Science, 664033 Irkutsk, Russia; 2Faculty of Physics, Irkutsk State University, 664003 Irkutsk, Russia; 3Laboratory of Quantum Chemical Modeling of Molecular Systems, Irkutsk State University, 664003 Irkutsk, Russia

**Keywords:** acylethynylpyrroles, alkynones, terminal alkynes, deacylation, *retro*-Favorsky reaction

## Abstract

It has been found that the addition of CH_2_CN^−^ anion to the carbonyl group of acylethynylpyrroles, generated from acetonitrile and *t*-BuOK, results in the formation of acetylenic alcohols, which undergo unexpectedly easy (room temperature) decomposition to ethynylpyrroles and cyanomethylphenylketones (*retro*-Favorsky reaction). This finding allows a robust synthesis of ethynylpyrroles in up to 95% yields to be developed. Since acylethynylpyrroles became available, the strategy thus found makes ethynylpyrroles more accessible than earlier. The quantum-chemical calculations (B2PLYP/6-311G**//B3LYP/6-311G**+C-PCM/acetonitrile) confirm the thermodynamic preference of the decomposition of the intermediate acetylenic alcohols to free ethynylpyrroles rather than their potassium derivatives.

## 1. Introduction

Ethynylpyrroles are valuable building blocks in the synthesis of many natural and synthetic biologically active compounds, such as antibiotic roseophilin, a potent cytotoxic agent against K562 human erythroid leukemia cells [1] and alkaloid quinolactacide with insecticidal activity [2]. They are applied in the syntheses of inhibitors of EGFR tyrosine kinase, an important target for anticancer drug design [3], the HMG-CoA reductase inhibitors for the treatment of hypercholesterolemia, hyperlipoproteinemia, hyperlipidemia and atherosclerosis [4], selective dopamine D4 receptor ligands [5] and foldamers, synthetic receptors, modified for encapsulation of dihydrogenphosphate ions [6]. Pyrroles with terminal acetylenic substituents take part in the syntheses of both lipophilic and highly hydrophilic BODIPY dyes, which fluoresce with high quantum yields and have low cytotoxicity, which makes it possible to visualize cells [7].

These pyrroles are employed in the development of advanced materials capable of detecting various organic and inorganic targets, such as tetrahedral oxoanions (H_2_PO_4_^−^ and SO_4_^2−^) [8] and pyrophosphate anions [9].

Also, high-tech materials, including ultrasensitive fluorescent probes for glucopyranoside [10], photoswitchable materials [11,12,13], components of dye-sensitized solar cells [14], monomers for organic thin-film transistors [15], prospective for energy storage devices, electrochemically active photoluminescence films are based on terminal ethynylpyrroles [16].

In light of the previous, it is clear that the improvement of the synthesis of ethynylpyrroles is a challenge. Indeed, the approaches to the preparation of these functionalized pyrroles are mainly limited to the deprotection of substituted at the triple bond (usually with TMS/TIPS groups) ethynylpyrroles, the products of the reaction of halopyrroles with the corresponding terminal acetylenes (Sonogashira cross-coupling) [2,5,6,8,17,18,19]. However, in this case, this coupling has limitations, since many halogenated pyrroles, except for representatives with electron-withdrawing substituents, are neither readily available nor stable [20,21]. Variants of the cross-coupling, such as Negishi reaction of halopyrroles with ethynyl magnesium chloride or zinc bromide [22] or cross-coupling of (1-methylpyrrol-2-yl)lithium with fluoroacetylene [23], are used albeit less often. It should be especially emphasized that almost all ethynylpyrroles synthesized by the above methods lack the substituents at carbon atoms in the pyrrole ring, i.e., the assortment of accessible ethynylpyrrole remains small and need to be extended.

Among other methods are Corey–Fuchs reaction of pyrrole-2-carbaldehydes with CBr_4_ with further conversion of dibromoolefins to ethynylpyrroles under the action of bases [1,3,7,24,25] and flash vacuum pyrolysis (FVP) of cyclic and linear 2-alkenylpyrroles (750 °C), limited to a few examples [26,27,28,29] due to difficulties in hardware implementation and requirements for substrates. Base-catalyzed elimination of ketones from tertiary acetylenic alcohols (*retro*-Favorsky reaction), affording pyrroles with terminal acetylenic substituents [30], is a rarer approach to such acetylenes because they could decompose or polymerize at high temperatures (up to 180 °C) common for the realization of this synthesis.

The formation of ethynylpyrroles as a result of the deacylation of acylethynylpyrroles was mentioned in only a few cases [31,32], and their yield was insignificant (though alkynones without pyrrole substituents in the presence of alkali metal hydroxides undergo hydrolytic cleavage to form terminal acetylenes [33,34,35]). For instance, when benzoylethynylpyrrole was treated with NaOH in DMSO (45–50 °C, 4 h), debenzoylation was detected by ^1^H NMR in negligible extent [31] and 7-days keeping of trifluroacetyl ethynylpyrrole over Al_2_O_3_ led to the ethynylpyrrole in 24% yield [32]. Certainly, these results were not suitable for the preparative synthesis of ethynylpyrroles.

Recently [36], we have disclosed the reaction of acylethynylpyrroles **1** with MeCN and metal lithium affording pyrrolyl-cyanopyridines **2** in up 87% yield (Figure 1).

The synthesis was accompanied by the formation of propargyl alcohols **3** (up to 15%) and small amounts of ethynylpyrroles **4** (up to 5%). The propargyl alcohols **3** were proved to be intermediates in the synthesis of both pyridines and ethynylpyrroles.

These results served as a clue to develop a novel synthesis of ethynylpyrroles, provided we could manage to turn the above side process into a major reaction. Our further successful experiments confirmed this assumption. It appeared that if lithium metal is replaced by *t*-BuOK, the reaction is shifted almost completely to the formation of side ethynylpyrroles. The progress of this synthesis optimization is illustrated in Table 1, wherein the most representative results are presented. As a reference compound, 3-(1-benzyl-4,5,6,7-tetrahydro-1*H*-indol-2-yl)-1-(thiophen-2-yl)prop-2-yn-1-one (**1a**), was chosen believing that the optimal conditions, found for this pyrrolyl acetylenic ketone of higher complexity, will also be valid for the simpler congeners.

In this paper, we report the exceptionally mild decarbonylation of acylethynylpyrroles, readily available from the reaction of pyrroles with electrophilic haloacetylenes in the medium of solid oxides and metal salts [32,37,38,39,40], under the action of CH_2_CN^−^ anion generated in situ in the system MeCN/*t*-BuOK.

## 2. Results and Discussion

As seen from Table 1, when the reaction was carried out by stirring acylethynylpyrrole **1a** with 2 eq. *n*-BuLi in MeCN at room temperature under ^1^H NMR control, the isolated crude product contained 12% of the target ethynylpyrrole **4a** (Table 1, Entry 1), i.e., the expected decarbonylation degree was noticeably increased. The major product, in this case, became tertiary propargylic alcohol **3a** (content in the reaction mixture was 78%). Pyrrolylpyridine **2a**, previously a major product [36], was also present in the reaction mixture but in a much smaller amount (10%). Almost the same results were obtained in the presence of 2 eq. of *t*-BuONa (Entry 3). But *t*-BuOLi turned out to be completely inactive in this reaction (Entry 2): the starting acylethynylpyrrole **1a,** in this case, was almost returned from the reaction.

*t*-BuOK catalyzed the formation of ethynylpyrrole **4a** much more actively: in the crude product obtained with one equivalent of this base, the content of the ethynylpyrrole **4a** in the reaction mixture attained 66% (Entry 4). However, under these conditions, the conversion of the starting acylethynylpyrrole **1a** was only 82%, but the content of pyrrolylpyridine **2a** in the reaction mixture increased to 16%.

When 2 eq. *t*-BuOK were used, acylethynylpyrrole **1a** reacted completely during the same time, and the content of ethynylpyrrole **4a** in the reaction mixture became 90%. Pyrrolylpyridine **2a** was also present as a by-product (10%) in the reaction mixture (Entry 5).

We found that it was possible to get rid of the pyridine almost completely (Entry 6 and 7) by carrying out the reaction in the mixed solvents (MeCN/THF or MeCN/DMSO in volume ratio 1:1). Thus, under these conditions, the reaction was excellently selective providing ethynylpyrrole **4a** in ~80% isolated yield.

**Table 1 molecules-28-01389-t001:** Optimization of the ethynylpyrrole **4a** synthesis by decarbonylation of acylethynylpyrrole **1a** ^a^.

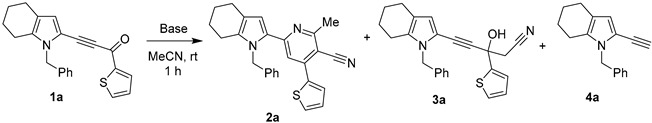
Entry	Base, eq.	Content in the crude, % (^1^H NMR)
1a	2a	3a	4a
1	*n*-BuLi, 2	traces	10	78	12
2	*t*-BuOLi, 2	~100	traces	traces	traces
3	*t*-BuONa, 2	traces	traces	85	15
4	*t*-BuOK, 1	18	16	traces	66
5	*t*-BuOK, 2	traces	10	traces	90
6 ^b^	*t*-BuOK, 2	traces	traces	traces	~100 ^d^
7 ^c^	*t*-BuOK, 2	traces	traces	traces	~100 ^e^

^a^—reaction conditions: 0.5 mmol of **1a**, acetonitrile (2.0 mL), 20–25 °C, nitrogen atmosphere. ^b^—the reaction was carried out in the MeCN/THF (1:1) system. ^c^—the reaction was carried out in the MeCN/DMSO (1:1) system. ^d^—isolated yield 84%. ^e^—isolated yield 82%.

Next, with the optimized reaction conditions (2 eq of *t*-BuOK, THF/MeCN, room temperature, 1 h) in hand, we have evaluated the scope of this reaction using benzoyl-, furoyl-, and thenoylethynylpyrroles with alkyl, aryl and hetaryl substituents at 4(4,5)-positions and methyl, benzyl, and vinyl moieties at the nitrogen atom of the pyrrole ring. Eventually, the series of earlier unknown ethynylpyrroles **4a**–**k** were synthesized in good to excellent yields, the exception being pyrrole **4d** (yield 36%) (Figure 2).

The method proved to be extendable over indole compounds, as shown in the example of 3-benzoylethynylindole **5**, which was transformed to the expected 1-methyl-3-ethynylindole **6** under the same conditions (Figure 3).

Thus, this result shows that 3-ethynylindoles—valuable synthetic building blocks [41]—could be more accessible than previously due to the above-elaborated strategy. Noteworthy that the starting 3-acylethynylindoles can be easily prepared by the cross-coupling of the corresponding *N*-substituted indoles with acylbromoacetylenes in solid Al_2_O_3_ media [42].

Also, we have attempted to extend the synthesis of ethynylpyrroles over the furan series. For this, we have chosen menthofuran, a natural antioxidant component of peppermint oil [43]. It turned out that 2-benzoylethynylmenthofuran **7**, which synthesis was previously described in [44], underwent similar decarbonylation under the above conditions to give the expected ethynyl derivative **8** in 80% yield (Figure 4).

The narrow range of the yields (74–95%) evidences that the structural effects on the synthesis efficiency are insignificant that are likely to result from the complex character of the process: (i) the formation of intermediate propargyl alcohols **3** and (ii) the decomposition of the latter. Besides, these steps are parallel to the formation of pyridine **2**. Apart from these competing factors, the yields are influenced by the isolation procedure (chromatography on the SiO_2_), wherein a noticeable amount of the target products are lost (Table 1, cf. ^1^H NMR and isolated yields). Nevertheless, the following general trend in yields may be noted: alkyl substituents in the pyrrole ring slightly decrease the reaction efficiency compared to aromatic substituents (74–86% vs. 84–95%). That can be referred to as a higher acidophobicity of the alkyl pyrroles.

It is known that MeCN is easily deprotonated by the action of alkali metals to give acetonitrile dimers via the formation of an intermediate CH_2_CN^−^ anion [45]. Also, it was reported that CH_2_CN^−^ anion was added to ketones to form tertiary cyanomethyl alcohols [46,47,48,49]. Correspondingly, in the previous communication, we have shown that the intermediate propargyl alcohol **3** are actually adducts of acylethynylpyrroles and CH_2_CN^−^ anion [36].

Although we failed to isolate propargyl alcohol **3a** in the reaction mixture obtained in the presence of *t*-BuOK, the results produced with *t*-BuONa allowed us to assume that in the first case, the reaction also proceeded with the formation of the intermediate **3a**, which was rapidly decomposed.

To verify this assumption, propargyl alcohols **3a**,**c**,**d**,**f**, prepared from acylethynylpyrroles **1a**,**c**,**d**,**f** and acetonitrile in the presence of *t*-BuONa according to the modified protocol (Figure 5) [36], were rapidly and quantitatively converted in the presence of *t*-BuOK into the corresponding ethynylpyrroles **4a**,**c**,**d**,**f** (Figure 5).

We performed the reaction in an NMR tube in deuterated acetonitrile. Immediate transformation of the characteristic signals of the protons of the benzoyl group at 8.16 ppm to protons of Ph-substituent occurs after the addition of *t*-BuOK to acylethynylpyrrole **1c** solution, which corresponds to the formation of the intermediate acetylenic alcohol **3c** (Figure 3). Additionally, two nearly equal singlets at 6.30 (signal of H-3 of pyrrole ring in ethynylpyrrole **4c**) and 6.44 ppm (signal of H-3 of pyrrole ring in intermediate alcohol **3c**) appeared. The singlet at 6.44 ppm decreases rapidly and disappears after about 30 min of reaction. After 1 h reaction mixture contained only terminal alkyne **4c** with a fully deuterated terminal acetylene position. Thus, the results confirm the proposed mechanism of the formation of ethynylpyrroles via intermediate acetylenic alcohol decomposition.

Cyanomethyl ketone (on the example of cyanomethyl-(2-thienyl)ketone **9a**), a second product of the *retro*-Favorsky reaction, was detected (^1^H NMR) after acidification of the aqueous suspension received during the workup of the reaction mixture (Figure 6).

Therefore, it is rigorously confirmed that in this reaction, ethynylpyrroles are the products of tertiary propargyl alcohol **3** decomposition, the *retro*-Favorsky reaction, which in this case occurs under extraordinarily mild (room temperature) conditions. Commonly this reaction requires a considerably higher temperature [120–140 °C (1 mm)] [30].

It could be emphasized that tertiary propargyl alcohols are one of the most attractive synthetic building blocks in organic synthesis [50,51,52,53,54,55,56,57,58]. This is primarily due to their bifunctionality (acetylene and hydroxyl functions), owing to which they can undergo cascade or multistage reactions with the formation of diverse compounds. In recent years, owing to the development of efficient methods for the synthesis of enantiomerically pure tertiary propargyl alcohols [59,60,61], interest in this class of compounds has increased significantly.

The tertiary propargyl alcohols here synthesized additionally contain one more synthetically valuable functional group (CN group and active C-H bond adjusted to nitrile function) and a pyrrole ring that significantly expands their potential for the design of novel functionalized compounds.

Despite the experimental evidence highlighting the mechanism of the cascade reaction studied, several mechanistic issues still need a quantum-chemical analysis. These issues mainly relate to the key stage of the synthesis, i.e., the *t*-BuOK-catalyzed decomposition of the intermediate propargyl alcohols **3**. Here the following questions should be clarified: (i) are the intermediates **3** decompose to the corresponding ketones **9** and potassium derivatives **11** of ethynylpyrroles as so far usually considered or free ethynylpyrroles **4** and the corresponding potassium enolates **10** (Figure 7) are formed? Although the Favorsky *retro*-reaction was synthetically thoroughly studied, this issue was never specially investigated. (ii) Is the experimentally observed formation of enolate from propargyl alcohols kinetically or thermodynamically controlled? (iii) Is the experimentally observed role of alkali metal cation, which fully controls the synthesis direction, an intrinsic (intramolecular) feature of the reaction, or is this influence of intermolecular solvation of the cations? (iv) What is the contribution of the solvent effect to the thermodynamics of this reaction?

To gain a clearer understanding of these mechanistic points, we have performed the quantum chemical calculations of the fundamental characteristics of the above reaction, the Gibbs free energy change, ∆G, using the DFT-based computational approach, which can be briefly referred to as B2PLYP/6-311G**//B3LYP/6-311G**+C-PCM/acetonitrile (see Appendix A for details) and assuming R^1^ = Me, R^2^ = R^3^ = H, R^4^ = Ph in Figure 5.

According to the results obtained, path A, i.e., formation of the metallated ethynylpyrroles and ketones (Figure 5), is thermodynamically closed, whereas path B (Figure 5), i.e., formation of the ethynylpyrrole and enolate, is thermodynamically opened (see SI for details). The calculations indicate that the decomposition of intermediate **3** proceeds via the formation of free ethynylpyrrole and potassium enolate. Also, these results evidence that path B is thermodynamically controlled. The ∆G values for path B calculated for Li, Na and K derivatives of propargyl alcohols **3** are −66.5, −78.6 and −88.3 kJ/mol, respectively. This explains experimental results according to which with the *t*-BuOLi, no products are formed (Table 1), while *t*-BuONa promotes the formation of sodium enolate, however stable under reaction conditions, and with *t*-BuOK, the decomposition of potassium enolate occurs. Thus, the effect of potassium alkali metal indeed has an intrinsic (intramolecular) character.

The computed O-Li, O-Na and O-K bond lengths in alkali metal derivatives of propargyl alcohols are 1.71, 2.08, and 2.43 Å, respectively, and in the corresponding enolates are 1.83, 2.16, 2.61 Å, respectively. These values correlate with the literature data: 1.95 Å (O-Li), 2.14–2.32 Å (O-Na), 2.60–2.80 Å (O-K) [62], respectively. The reported bond energies are 343, 255 and 238 kJ/mol [63]. From these results, it becomes clear why with *t*-BuOK, the pyridines **2** are not formed: the abstraction of a proton from the CHCN moiety would lead to dianionic-like species that are thermodynamically unfavorable. In the cases of Li- and Na-derivatives of propargyl alcohols, the negative charges on oxygen are smaller since they are tighter ion pairs, especially with lithium cation. Therefore, the reaction takes other directions: with Li cation, expectedly, pyridines are formed, and with *t*-BuONa, the propargyl alcohol decomposition slows down (Table 1, Entry 3).

The mechanism of ethynylpyrroles formation from potassium derivatives of propargyl alcohols (on the example of alcoholate **12**, R^1^ = Me, R^2^ = R^3^ = H, R^4^ = Ph) likely represents an intramolecular process (Figure 8) [36], involving the C_sp_-CH bond cleavage with simultaneous transfer of a proton from the CH bond.

This process is probably facilitated by the intramolecular interaction (coordination) between potassium cation and CN-bond (intermediate **A**). This is supported by the fact that the calculated K⋯N distance in the potassium derivative of propargyl alcohol (3.87 or 4.12 Å, depending on the molecular conformation, see Appendix A) is smaller than the sum of the van der Waals radii of these atoms (4.2 Å). The above-mentioned two conformations are separated only by 0.8 kJ/mol. Since the latter value is well within the error margin of our computational scheme, they both can be considered legitimate propargyl alcohol equilibrium ground-state molecular structures (see Appendix A for more details).

The ∆G values computed for the formation of ethynylpyrroles with the participation of the solvent (MeCN) and then without (gas phase) are close (−88.3 and −84.5 kJ/mol). This means that the contribution of the solvent effect is negligible.

The experiments with MeNO_2_ showed that in this solvent, the reaction did not proceed at all: the starting acylethynylpyrrole was recovered completely. In our previous work [36], we reported the reaction of benzoylethynylpyrrole **1a** with isobutyronitrile and valeronitrile in the presence of lithium metal. In both cases, respective intermediate alcohols were isolated in 60 and 26% yields. In the presence of *t*-BuOK, both were readily transformed to corresponding ethynylpyrrole **4a**.

## 3. Experimental Section

### 3.1. General Information

IR spectra were obtained on a “Bruker IFS-25” spectrometer (Bruker, Billerica, MA, USA) (KBr pellets or films in 400–4000 cm^−1^ region). ^1^H (400.13 MHz) and ^13^C (100.6 MHz) NMR spectra were recorded on a “Bruker Avance 400” instrument (Bruker, Billerica, MA, USA) in CDCl_3_. The assignment of signals in the ^1^H NMR spectra was made using COSY and NOESY experiments. Resonance signals of carbon atoms were assigned based on ^1^H-^13^C HSQC and ^1^H-^13^C HMBC experiments. The ^1^H chemical shifts (δ) were referenced to the residual solvent protons (7.26 ppm, CDCl_3_), and the ^13^C chemical shifts were expressed with respect to the deuterated solvent (77.16 ppm). Coupling constants in hertz (Hz) were measured from one-dimensional spectra, and multiplicities were abbreviated as follows: br (broad), s (singlet), d (doublet), t (triplet), and m (multiplet). The chemical shifts were recorded in ppm. The (C, H, N) microanalyses were performed on a Flash EA 1112 CHNS-O/MAS (CHN Analyzer) instrument (Thermo Finnigan, Italy). Sulfur was determined by complexometric titration with Chlorasenazo III. Fluorine content was determined on a SPECOL 11 (Carl Zeiss Jena, Germany) spectrophotometer. Melting points (uncorrected) were determined with SMP50 Stuart Automatic melting point (Cole-Palmer Ltd. Stone, Staffordshire, UK).

### 3.2. Synthesis of Ethynylpyrroles ***4a**–**k***, Ethynylindole ***6***, Ethynylfuran ***8***, General Procedure

Acylethynylpyrrole **1a**–**k**, 3-acylethynylindole **5** or 2-acylethynylfuran **7** (1 mmol) was dissolved in dry THF/MeCN (1:1, 4 mL), and then *t*-BuOK (224 mg, 2 mmol) was added to reaction mixture under nitrogen. Reaction mixture was stirred at room temperature for 1 h while turning into an orange suspension. Then reaction mixture was diluted with cold (0–5 °C) water (30 mL) and extracted by cold (0–5 °C) *n*-hexane (3 × 10 mL). Combined extracts were washed with water (3 × 5 mL) and dried over Na_2_SO_4_. The residue, after removing solvent, was purified by flash chromatography (dried SiO_2_, *n*-hexane) to afford ethynylpyrrole **4a**–**k**, ethynylindole **6** and ethynylfuran **8**.

*1-Benzyl-2-ethynyl-4,5,6,7-tetrahydro-1H-indole* (**4a**). Yield: 197 mg (84%), colorless oil; ^1^H NMR (400.13 MHz, CDCl_3_): δ 7.37–7.24 (m, 3H, H*m,p,* Ph), 7.13–7.08 (m, 2H, H*o,* Ph), 6.37 (s, 1H, H-3, pyrrole), 5.14 (s, 2H, CH_2_-Ph), 3.35 (s, 1H, ≡CH), 2.54–2.49 (m, 2H, CH_2_-7), 2.43–2.38 (m, 2H, CH_2_-4), 1.82–1.68 (m, 4H, CH_2_-5, CH_2_-6); ^13^C NMR (100.6 MHz, CDCl_3_): δ 138.3, 130.9, 128.7 (2C), 127.3, 126.7 (2C), 118.0, 114.2, 99.7, 81.2, 77.0, 47.9, 23.5, 23.2, 23.1, 22.5; IR (KBr) 3287, 3087, 3063, 3030, 2928, 2849, 2097, 1495, 1457, 1388, 1357, 1301, 1130, 1077, 1029, 928, 795, 722, 696, 545, 457 cm^−1^; Anal. Calcd for C_17_H_17_N: C, 86.77; H, 7.28; N, 5.95%. Found: C, 86.47; H, 7.31; N, 6.14%.

*2-Ethynyl-1-methyl-4,5,6,7-tetrahydro-1H-indole* (**4b**). Yield: 137 mg (86%), white crystals, mp 53–54 °C; ^1^H NMR (400.13 MHz, CDCl_3_): δ 6.26 (s, 1H, H-3, pyrrole), 3.50 (s, 3H, NMe), 3.37 (s, 1H, ≡CH), 2.52–2.50 (m, 2H, CH_2_-7), 2.47–2.45 (m, 2H, CH_2_-4), 1.83–1.80 (m, 2H, CH_2_-5), 1.73–1.71 (m, 2H, CH_2_-6); ^13^C NMR (100.6 MHz, CDCl_3_,): δ 130.9, 117.4, 113.6, 112.6, 81.1, 76.9, 30.8, 23.6, 23.2, 23.0, 22.4; IR (film) 3288, 3100, 2929, 2847, 2097, 1570, 1462, 1442, 1386, 1302, 1130, 1055, 790, 667, 536 cm^−1^; Anal. Calcd for C_11_H_13_N: C, 82.97; H, 8.23; N, 8.80%. Found: C, 82.71; H, 8.44; N, 8.58%.

*2-Ethynyl-1-vinyl-4,5,6,7-tetrahydro-1H-indole* (**4c**). Yield: 127 mg (74%), colorless oil; ^1^H NMR (400.13 MHz, CDCl_3_): δ 6.97 (dd, *J* = 16.1, 9.4 Hz, 1H, H_x_), 6.34 (s, 1H, H-3, pyrrole), 5.34 (d, *J* = 16.1 Hz, 1H, H_a_), 4.83 (d, *J* = 9.4 Hz, 1H, H_b_), 3.39 (s, 1H, ≡CH), 2.66–2.63 (m, 2H, CH_2_-7), 2.48–2.45 (m, 2H, CH_2_-4), 1.83–1.80 (m, 2H, CH_2_-5), 1.71–1.69 (m, 2H, CH_2_-6); ^13^C NMR (100.6 MHz, CDCl_3_): δ 130.5, 119.5, 116.8, 112.3, 102.1, 99.7, 82.1, 76.8, 24.2, 23.4, 23.1, 23.0; IR (film) 3292, 3128, 3049, 2932, 2849, 2099, 1643, 1577, 1483, 1438, 1387, 1324, 1294, 1136, 966, 871, 802, 669, 558 cm^−1^; Anal. Calcd for C_12_H_13_N: C, 84.17; H, 7.65; N, 8.18%. Found: C, 83.85; H, 7.81; N, 8.36%.

*5-Ethynyl-2,3-dimethyl-1-vinyl-1H-pyrrole* (**4d**). Yield: 52 mg (36%), colorless oil; ^1^H NMR (400.13 MHz, CDCl_3_): δ 6.91 (dd, *J* = 16.0, 9.2 Hz, 1H, H_x_), 6.36 (s, 1H, H-3, pyrrole), 5.46 (d, *J* = 16.1 Hz, 1H, H_a_), 4.94 (d, *J* = 9.2 Hz, 1H, H_b_), 3.37 (s, 1H, ≡CH), 2.21 (s, 3H, Me), 1.99 (s, 3H, Me); ^13^C NMR (100.6 MHz, CDCl_3_): δ 130.8, 127.8, 119.0, 116.7, 111.6, 104.5, 81.7, 76.9, 11.4, 11.1; IR (film) 3291, 3106, 2920, 2866, 2099, 1643, 1483, 1432, 1392, 1335, 1310, 1162, 1113, 965, 879, 806, 671, 562 cm^−1^; Anal. Calcd for C_10_H_11_N: C, 82.72; H, 7.64; N, 9.65%. Found: C, 82.94; H, 7.49; N, 9.80%.

*2-Ethynyl-1-methyl-5-phenyl-1H-pyrrole* (**4e**). Yield: 172 mg (95%), colorless oil; ^1^H NMR (400.13 MHz, CDCl_3_): δ 7.42–7.34 (m, 5H, Ph), 6.55 (d, *J* = 3.8 Hz, 1H, H-3, pyrrole), 6.16 (d, *J* = 3.8 Hz, 1H, H-4, pyrrole), 3.69 (s, 3H, NMe), 3.44 (s, 1H, ≡CH); ^13^C NMR (100.6 MHz, CDCl_3_): δ 136.7, 132.9, 128.9 (2C), 128.6 (2C), 127.5, 116.1, 115.6, 108.6, 82.0, 76.5, 33.2; IR (film) 3287, 3106, 3060, 2948, 2102, 1602, 1498, 1457, 1390, 1324, 1234, 1155, 1074, 1028, 758, 698, 568 cm^−1^; Anal. Calcd for C_13_H_11_N: C, 86.15; H, 6.12; N, 7.73%. Found: C, 85.75; H, 5.86; N, 7.48%.

*2-Ethynyl-5-(4-methylphenyl)-1-vinyl-1H-pyrrole* (**4f**). Yield: 174 mg (84%), colorless oil; ^1^H NMR (400.13 MHz, CDCl_3_): δ 7.34–7.28 (m, 2H, H*o,* Ph), 7.24–7.17 (m, 2H, H*m,* Ph), 6.82 (dd, *J* = 15.9, 9.0 Hz, 1H, H_x_), 6.63 (d, *J* = 3.8 Hz, 1H, H-3 pyrrole), 6.17 (d, *J* = 3.8 Hz, 1H, H-4, pyrrole), 5.53 (d, *J* = 15.9 Hz, 1H, H_a_), 4.99 (d, *J* = 9.0 Hz, 1H, H_b_), 3.43 (s, 1H, ≡CH), 2.38 (s, 3H, Me); ^13^C NMR (100.6 MHz, CDCl_3_): δ 137.6, 136.1, 131.1, 129.6, 129.2 (2C), 129.1 (2C), 118.5, 114.5, 109.9, 107.0, 82.5, 76.8, 21.3; IR (KBr) 3287, 3112, 3024, 2921, 2102, 1643, 1547, 1510, 1466, 1419, 1389, 1324, 1297, 1226, 1113, 963, 889, 822, 775, 672, 571, 500 cm^−1^; Anal. Calcd for C_15_H_13_N: C, 86.92; H, 6.32; N, 6.76%. Found: C, 86.68; H, 6.51; N, 6.85%.

*1-Benzyl-2-ethynyl-5-(4-methoxyphenyl)-1H-pyrrole* (**4g**). Yield: 253 mg (88%), white crystals; mp 92–93 °C; ^1^H NMR (400.13 MHz, CDCl_3_): δ 7.30–7.22 (m, 3H, H*m,p,* Ph), 7.20–7.15 (m, 2H, H*o,* Ph), 6.99–6.93 (m, 2H, H*m,* Ph), 6.87–6.82 (m, 2H, H*o,* Ph), 6.63 (d, *J* = 3.7 Hz, 1H, H-3 pyrrole), 6.16 (d, *J* = 3.7 Hz, 1H, H-4, pyrrole), 5.25 (s, 2H, CH_2_-Ph), 3.80 (s, 3H, MeO), 3.29 (s, 1H, ≡CH); ^13^C NMR (CDCl_3_, 100.6 MHz): δ 159.3, 138.8, 136.7, 130.4 (2C), 128.6 (2C), 127.2, 126.3 (2C), 125.3, 116.1, 115.5, 114.0 (2C), 108.8, 81.8, 76.6, 55.4, 48.9; IR (KBr) 3287, 3087, 3063, 3031, 2955, 2934, 2836, 2100, 1611, 1575, 1547, 1510, 1463, 1442, 1392, 1358, 1321, 1288, 1249, 1178, 1110, 1087, 1031, 977, 909, 836, 767, 731, 695, 575, 524, 459 cm^−1^; Anal. Calcd for C_20_H_17_NO: C, 83.59; H, 5.96; N, 4.87; O, 5.57%. Found: C, 83.31; H, 6.02; N, 5.02%.

*2-Ethynyl-5-(2-fluorophenyl)-1-vinyl-1H-pyrrole* (**4h**). Yield: 192 mg (91%), colorless oil; ^1^H NMR (400.13 MHz, CDCl_3_): δ 7.40–7.30 (m, 2H, H*m,* Ph), 7.22–7.08 (m, 2H, H*o,p,* Ph), 6.84 (dd, *J* = 15.9, 8.9 Hz, 1H, H_x_), 6.66 (d, *J* = 3.7 Hz, 1H, H-3 pyrrole), 6.24 (d, *J* = 3.7 Hz, 1H, H-4, pyrrole), 5.34 (d, *J* = 15.9 Hz, 1H, H_a_), 4.91 (d, *J* = 8.9 Hz, 1H, H_b_), 3.45 (s, 1H, ≡CH); ^13^C NMR (100.6 MHz, CDCl_3_): δ 159.9 (d, *J* = 249.1 Hz, C-2, 2-FC_6_H_4_), 132.1 (d, *J* = 2.0 Hz, C-6, 2-FC_6_H_4_), 130.9, 130.1 (d, *J* = 8.2 Hz, C-4, 2-FC_6_H_4_), 129.2, 124.24 (d, *J* = 3.3 Hz, C-5, 2-FC_6_H_4_), 120.6 (d, *J* = 15.5 Hz, C-1, 2-FC_6_H_4_), 118.1, 116.1 (d, *J* = 22.0 Hz, C-3, 2-FC_6_H_4_), 115.2, 111.8, 106.4, 82.7, 76.4; IR (KBr) 3293, 3115, 3068, 2924, 2104, 1645, 1580, 1547, 1498, 1465, 1397, 1300, 1229, 1109, 963, 890, 817, 780, 759, 672, 577, 471 cm^−1^; Anal. Calcd for C_14_H_10_FN: C, 79.60; H, 4.77; F, 8.99; N, 6.63%. Found: C, 79.24; H, 4.96; F, 8.75; N, 6.39%.

*5-Ethynyl-2,3-diphenyl-1-vinyl-1H-pyrrole* (**4i**). Yield: 242 mg (90%), white crystals; mp 93–94 °C; ^1^H NMR (400.13 MHz, CDCl_3_): δ 7.39–7.34 (m, 3H, H*o,p,* Ph), 7.31–7.26 (m, 2H, H*o,* Ph), 7.21–7.15 (m, 2H, H*m,* Ph), 7.15–7.09 (m, 3H, H*m,p,* Ph), 6.84 (s, 1H, H-3 pyrrole), 6.71 (dd, *J* = 15.9, 9.2 Hz, 1H, H_x_), 5.47 (d, *J* = 15.9 Hz, 1H, H_a_), 4.91 (d, *J* = 9.2 Hz, 1H, H_b_), 3.46 (s, 1H, ≡CH); ^13^C NMR (100.6 MHz, CDCl_3_): δ 135.1, 131.9, 131.8, 131.4 (2C), 130.8, 128.7 (2C), 128.3 (2C), 128.2 (3C), 126.1, 123.8, 118.6, 113.7, 106.5, 82.8, 76.5; IR (KBr) 3274, 3080, 3057, 2923, 2100, 1641, 1601, 1557, 1495, 1446, 1386, 1320, 1305, 1177, 1031, 964, 889, 800, 769, 699, 587, 522 cm^−1^; Anal. Calcd for C_20_H_15_N: C, 89.19; H, 5.61; N, 5.20%. Found: C, 88.89; H, 5.45; N, 5.34%.

*2-Ethynyl-1-methyl-5-(thiophen-2-yl)-1H-pyrrole* (**4j**). Yield: 174 mg (93%), colorless oil; ^1^H NMR (400.13 MHz, CDCl_3_): δ 7.32–7.28 (m, 1H, H-5, thiophene), 7.10–7.05 (m, 2H, H-3,4, thiophene), 6.51 (d, *J* = 3.9 Hz, 1H, H-3 pyrrole), 6.26 (d, *J* = 3.9 Hz, 1H, H-4, pyrrole), 3.76 (s, 3H, N-CH_3_), 3.43 (s, 1H, ≡CH); ^13^C NMR (100.6 MHz, CDCl_3_): δ 134.4, 129.2, 127.5, 125.8, 125.3, 116.6, 115.6, 109.7, 99.7, 82.2, 33.2; IR (KBr) 3288, 3106, 3074, 2944, 2922, 2101, 1445, 1417, 1395, 1345, 1314, 1201, 1034, 845, 766, 698, 570, 493 cm^−1^; Anal. Calcd for C_11_H_9_NS: C, 70.55; H, 4.84; N, 7.48; S, 17.12%. Found: C, 70.26; H, 4.69; N, 7.28; S, 16.82%.

*1-Benzyl-2-ethynyl-1H-pyrrole* (**4k**). Yield: 145 mg (80%), colorless oil; ^1^H NMR (400.13 MHz, CDCl_3_): δ 7.36–7.27 (m, 3H, H*m,p,* Ph), 7.16–7.14 (m, 2H, H*o,* Ph), 6.68–6.65 (m, 1H, H-3, pyrrole), 6.54–6.51 (m, 1H, H-5, pyrrole), 6.13–6.10 (m, 1H, H-4, pyrrole), 5.19 (s, 2H, CH_2_-Ph), 3.33 (s, 1H, ≡CH); ^13^C NMR (CDCl_3_, 100.6 MHz): δ 137.9, 128.8 (2C), 127.7, 127.3 (2C), 123.1, 116.0, 114.7, 108.7, 81.7, 76.0, 51.3; IR (KBr) 3288, 3106, 3064, 3031, 2925, 2853, 2103, 1495, 1466, 1455, 1435, 1300, 1018, 722, 694, 569, 522 cm^−1^; Anal. Calcd for C_13_H_11_N: C, 86.15; H, 6.12; N, 7.73%. Found: C, 85.84; H, 5.89; N, 7.45%.

*3-Ethynyl-1-methyl-1H-indole* (**6**). Yield: 113 mg (73%); Spectral characteristics are the same as previously published [64].

*2-Ethynyl-3,6-dimethyl-4,5,6,7-tetrahydrobenzofuran* (**8**). Yield: 139 mg (80%), colorless oil; ^1^H NMR (400.13 MHz, CDCl_3_): δ 3.55 (s, 1H, ≡CH), 2.67–2.62 (m, 1H, CH), 2.33–2.30 (m, 2H, CH_2_), 2.19–2.12 (m, 1H, CH), 2.00 (s, 3H, Me), 1.93–1.91 (m, 1H, CH), 1.85–1.81 (m, 1H, CH), 1.36–1.30 (m, 1H, CH), 1.07 (d, *J* = 6.7 Hz, 3H, CHMe); ^13^C NMR (100.6 MHz, CDCl_3_): δ 152.1, 131.5, 127.2, 118.4, 83.8, 74.7, 31.7, 31.2, 29.6, 21.5, 20.0, 9.0; IR (KBr) 3293, 2923, 2849, 2103, 1628, 1558, 1456, 1379, 1295, 1257, 1150, 1107, 1066, 1041, 774, 692 cm^−1^; Anal. Calcd for C_12_H_14_O: C, 82.72; H, 8.10; O, 9.18%. Found: C, 82.94; H, 7.88%.

### 3.3. Synthesis of Propargyl Alcohols ***3a**,**c**,**d**,**f***

Acylethynylpyrrole **1a**,**c**,**d**,**f** (1 mmol) was dissolved in dry MeCN (4 mL), and then *t*-BuONa (192 mg, 2 mmol) was added to reaction mixture under nitrogen and reaction mixture was stirred at room temperature for 1 h. Then reaction mixture was diluted with water (30 mL) and extracted by diethyl ether (3 × 10 mL). Extracts were washed with water (3 × 5 mL) and dried over Na_2_SO_4_. The residue after removing solvents was fractionated by column chromatography (SiO_2_, *n*-hexane:diethyl ether, 10:1) to afford propargyl alcohol **3a**,**c**,**d**,**f**.

*5-(1-Benzyl-4,5,6,7-tetrahydro-1H-indol-2-yl)-3-hydroxy-3-(thiophen-2-yl)pent-4-ynenitrile* (**3a**). Spectral characteristics are the same as previously published [36].

*3-Hydroxy-3-phenyl-5-(1-vinyl-4,5,6,7-tetrahydro-1H-indol-2-yl)pent-4-ynenitrile* (**3c**). Yield: 224 mg (71%), yellow oil; ^1^H NMR (400.13 MHz, CDCl_3_): δ 7.72–7.71 (m, 2H, H*o*, Ph), 7.44–7.37 (m, 3H, H*m,p*, Ph), 6.98 (dd, *J* = 15.9, 9.3 Hz, 1H, H_x_), 6.40 (s, 1H, H-3, pyrrole), 5.34 (d, *J* = 15.9 Hz, 1H, H_a_), 4.88 (d, *J* = 9.3 Hz, 1H, H_b_), 3.03 (d, *J* = 4.8 Hz, 2H, CH_2_CN), 2.85 (s, 1H, OH), 2.67–2.65 (m, 2H, CH_2_-7), 2.49–2.47 (m, 2H, CH_2_-4), 1.83–1.81 (m, 2H, CH_2_-5), 1.74–1.73 (m, 2H, CH_2_-6); ^13^C NMR (CDCl_3_, 100.6 MHz): δ 141.7, 131.5, 130.4, 129.0, 128.8 (2C), 125.4 (2C), 119.9, 117.3, 116.4, 111.3, 103.2, 92.9, 81.4, 71.0, 35.7, 24.1, 23.3, 23.1, 23.0. IR (film) 3422, 3062, 3030, 2931, 2851, 2215, 1643, 1492, 1447, 1383, 1295, 1241, 1143, 1102, 1053, 968, 910, 805, 765, 733, 700, 646 cm^−1^; Anal. Calcd for C_21_H_20_N_2_O: C, 79.72; H, 6.37; N, 8.85; O, 5.06%. Found: C, 79.44; H, 6.20; N, 8.59%.

*5-(4,5-Dimethyl-1-vinyl-1H-pyrrol-2-yl)-3-hydroxy-3-phenylpent-4-ynenitrile* (**3d**). Yield: 197 mg (68%), yellow crystals, mp 101–102 °C; ^1^H NMR (400.13 MHz, CDCl_3_): δ 7.72–7.70 (m, 2H, H*o*, Ph), 7.42–7.40 (m, 2H, H*m,p*, Ph), 6.91 (dd, *J* = 15.9, 9.1 Hz, 1H, H_x_), 6.41 (s, 1H, H-3, pyrrole), 5.45 (d, *J* = 15.9 Hz, 1H, H_a_), 4.99 (d, *J* = 9.1 Hz, 1H, H_b_), 3.02 (d, *J* = 5.1 Hz, 2H, CH_2_CN), 2.86 (s, 1H, OH), 2.22 (s, 3H, Me), 2.00 (s, 3H, Me); ^13^C NMR (100.6 MHz, CDCl_3_): δ 141.7, 130.6, 128.9, 128.7 (2C), 128.6, 125.4 (2C), 119.5, 117.0, 116.4, 110.6, 105.6, 92.6, 81.4, 70.9, 35.6, 11.3, 11.1; IR (KBr) 3422, 3062, 3030, 2921, 2215, 1643, 1493, 1449, 1392, 1357, 1304, 1172, 1100, 1049, 967, 910, 809, 765, 733, 700, 634 cm^−1^; Anal. Calcd for C_19_H_18_N_2_O: C, 78.59; H, 6.25; N, 9.65; O, 5.51%. Found: C, 78.22; H, 6.02; N, 9.42%.

*3-Hydroxy-3-phenyl-5-(5-(4-methylphenyl)-1-vinyl-1H-pyrrol-2-yl)pent-4-ynenitrile* (**3f**). Yield: 281 mg (80%), yellow oil; ^1^H NMR (CDCl_3_, 400 MHz): δ 7.74–7.72 (m, 2H, H*o*, Ph), 7.45–7.39 (m, 2H, H*m,p*, Ph), 7.31 (d, *J* = 7.9 Hz, 2H, H*o*, C_6_H_4_), 7.21 (d, *J* = 7.9 Hz, 2H, H*m*, C_6_H_4_), 6.83 (dd, *J* = 15.8, 8.9 Hz, 1H, H_x_), 6.67 (d, *J* = 3.8 Hz, 1H, H-4, pyrrole), 6.22 (d, *J* = 3.8 Hz, 1H, H-3, pyrrole), 5.52 (d, *J* = 15.8 Hz, 1H, H_a_), 5.05 (d, *J* = 8.9 Hz, 1H, H_b_), 3.06 (d, *J* = 5.0 Hz, 2H, CH_2_CN), 2.85 (s, 1H, OH), 2.39 (s, 3H, Me); ^13^C NMR (CDCl_3_, 100.6 MHz): δ 141.6, 137.8, 136.8, 131.2, 129.4, 129.3 (2C), 129.1 (3C), 128.8 (2C), 125.4 (2C), 118.9, 116.3, 113.7, 110.1, 108.0, 93.1, 81.4, 71.0, 35.6, 21.4; IR (KBr) 3416, 3061, 3028, 2922, 2218, 1643, 1515, 1472, 1449, 1418, 1389, 1324, 1301, 1224, 1112, 1042, 964, 909, 823, 773, 733,701, 622, 503 cm^−1^, Anal. Calcd for C_24_H_20_N_2_O: C, 81.79; H, 5.72; N, 7.95; O, 4.54%. Found: C, 81.35; H, 5.60; N, 7.68%.

## 4. Conclusions

In conclusion, we have found efficient and extraordinarily easy (room temperature) access to ethynylpyrroles via decarbonylation of available acylethynylpyrroles. The reaction proceeds in the MeCN-THF/*t*-BuOK system via the addition of CH_2_CN^−^ anion to the carbonyl group of acylethynylpyrroles followed by *retro*-Favorsky reaction of the intermediated propargylic alcohols. Thermodynamic aspects of the intermediate alcohol decomposition have been considered in the framework of B2PLYP/6-311G**//B3LYP/6-311G**+C-PCM/acetonitrile methodology. The substrate scope of the reaction includes benzoyl-, furoyl-, thenoylethynylpyrroles with alkyl, vinyl, aryl and hetaryl substituents at 1(4,5)-positions of the pyrrole ring, and methyl, benzyl, and vinyl moieties at the nitrogen atom, as well as acylethynyl derivatives of 1-methylindole and menthofuran.

## Data Availability

The data presented in this study are available in the Appendix A.

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
