# Peer review of "Unexpected Decarbonylation of Acylethynylpyrroles under the Action of Cyanomethyl Carbanion: A Robust Access to Ethynylpyrroles"

_molecules, 2023, doi:10.3390/molecules28031389_

Round 1

Reviewer 1 Report

This manuscript reports the synthesis of pyrroles with terminal acetylenic substituents from the decarbonylation of acylethynylpyrroles.

The author claimed that this approach will be an efficient and extraordinarily easy access to ethynylpyrroles and gave several examples exaggerating how difficult other methods would be to access ethynylpyrroles. However, the statement does not hold water at all. The efforts the authors paid for substrates i.e. acylethynylpyrroles synthesis should be similar and enough for ethynylpyrroles synthesis. I could imagine that the authors have to access their substrates via sonogahsira coupling or all the other methods the authors denounced. On the other hand, this approach lacks atom economy if someone really want to use this method.

Another problem with this manuscript is that this chemistry is not new. People have already reported similar transformations from acylethynylpyrroles under basic conditions to yield terminal alkynes (for example, reference 30, Izvestiya 347 Akademii Nauk SSSR, Seriya Khimicheskaya 1980, 8, 1346–1350.) Moreover, in the author’s previous publication (reference 36 New J. Chem. 2022, 46, 13149-13155.), the author has already shown they could get up to 63% yield of the decarbonylation product (ethynylpyrrole), while the final DFT calculations looks like a pure exercise in style.

In a nutshell the authors just demonstrated in the present paper that this already known transformation could undergo smoothly and fast using tBuOK/MeCN. To my opinion, this approach will be a helpful tool for late-stage modification if anybody wants to get ethynylpyrroles from existed acylethynylpyrroles in their inventory.

With all due respect to the authors, I cannot be supportive of this submission, but I can see how this work can be of interest to a very specialized audience. I suggest the authors rewrite the story and submit this work to other journals like Mendeleev commun. or Tetrahedron lett.

Author Response

  1. The author claimed that this approach will be an efficient and extraordinarily easy access to ethynylpyrroles and gave several examples exaggerating how difficult other methods would be to access ethynylpyrroles. However, the statement does not hold water at all. The efforts the authors paid for substrates i.e. acylethynylpyrroles synthesis should be similar and enough for ethynylpyrroles synthesis. I could imagine that the authors have to access their substrates via sonogahsira coupling or all the other methods the authors denounced. On the other hand, this approach lacks atom economy if someone really want to use this method.

Answer:

We understand the critics of the respected reviewer, which, in our opinion, is due to our overemphasizing the synthesis of ethynylpyrroles as a primary issue of the paper. Indeed, as follows from the paper title, first of all, we would like to draw the attention of the synthetic community to unexpectedly ready decarbonylation of acylethynylpyrroles, proceeding also via uncommonly easy decomposition of the intermediate acetylenic alcohol (retro-Favorsky reaction). The synthesis of ethynylpyrroles is considered by us just as a second, though practically important, output of the reaction found. This synthesis, to our knowledge, can be a valuable complementation to the existing routes providing the title compounds.

As far as atom economy is concerned, the method elaborated by us is quite competitive with knows ones, because the latter are either multi-step or require protection/deprotection procedures (Sonogashira or other couplings) that also compromises the atom economy. Besides, the Sonogashira coupling is not valid for ethynylation of pyrroles because the starting halogenated pyrroles are not stable.

Note that the starting acylethynylpyrroles are synthesized by the transition metal-free, room temperature cross-coupling of pyrroles with haloacetylenes. Now they are widely applied for the synthesis of heterocyclic compounds (see ref. 35, 37-40 given in the manuscript).

Nevertheless, we have tried to address the reviewer’s critics and correspondingly re-edited abstract, introduction and conclusion to quench the above overemphasizing.

  1. Another problem with this manuscript is that this chemistry is not new. People have already reported similar transformations from acylethynylpyrroles under basic conditions to yield terminal alkynes (for example, reference 30, Izvestiya 347 Akademii Nauk SSSR, Seriya Khimicheskaya 1980, 8, 1346–1350.) Moreover, in the author’s previous publication (reference 36 New J. Chem. 2022, 46, 13149-13155.), the author has already shown they could get up to 63% yield of the decarbonylation product (ethynylpyrrole), while the final DFT calculations looks like a pure exercise in style.

Answer: As said above the stem of the paper is uncommonly easy decarbonylation of acylethynylpyrroles, which is due to also uncommonly facile decomposition of the intermediate acetylenic alcohol (retro-Favorsky reaction) that is affected by cyanomethyl carbanion generated in situ from acetonitrile under the action of t-BuOK. Although a few examples of decarbonylation of acylethynylpyrroles in the presence of bases in small, non-preparative yields were mentioned in some publications, but those facts matter neither fundamentally nor practically.    It is worthwhile to underline that all the known methods for ethynylpyrroles synthesis were realized only on the examples of unsubstituted pyrroles, whereas this method was demonstrated to cover a quite a large number of ring-substituted pyrroles. Moreover, as it was complemented in the revised manuscript it was extendable to acylethynyl derivatives of indole and furan series. Therefore, the novelty of results presented in the paper should not be overlooked.     A special interest to our understanding is a novel insight into the mechanism of retro-Favorsky reaction based on quantum-chemical calculations, which allow for the first time to show clearly that acetylenes produced by this reaction are formed as terminal ≡CH species, but not as metal acetylides, as earlier considered. It seems to us that this result was adequately formulated in the abstract, manuscript and conclusion.              

Reviewer 2 Report

This manuscript describes an efficient access to ethynylpyrroles via decarbonylation of available acylethynylpyrroles. The reaction proceeds in the MeCN-THF/t-BuOK system via the addition of −CH2CN anion to the carbonyl group of acylethynylpyrroles followed by retro-Favorsky reaction of the intermediated propargylic alcohols. It is an unexpected and high-selective reaction. Its disadvantages are that the source of starting materials are not commercial available, and the  atom economy is low. However, I still think it is an interesting reaction, and it is suitable for publication in this journal.

Author Response

1.      Disadvantages are that the source of starting materials are not commercial available, and the  atom economy is low. Answer: Approaches to acylethynylpyrroles by reaction of pyrroles with haloacetylenes are referenced in the paper [35, 37-40]. Pyrroles are synthesized from cheap and commercially available ketones by Trofimov reaction by one-step procedure (10.13109/9783666351471.11). Haloacetylenes were synthesized by described procedure (https://doi.org/10.1007/s10593-013-1252-y) from aldehydes in multigram scale. 

Reviewer 3 Report

The presented manuscript can certainly be published, but requires minor modifications.

1. Abstract and conclusion are identical. I think the authors should rewrite the conclusion. Make it an independent part of the text.

2. There is a typo in the introduction, lines 36, 37 and 38.

3. The manuscript lacks an experimental part as such. The authors position their article as a research one. In this case, the instrument component and reagents used for research should be indicated. If this is a mini-review, then it must be supplemented with information.

4. In the section describing the reaction mechanism proposed by the authors (Scheme 6), the energy is given in kilocalories per mol. In units of the international system of units, energy should be considered in joules per mole of substance.

I think the authors will not be difficult to correct the comments made. The article certainly deserves to be published in the journal Molecules.

Recommendation: Accept with Minor revision.

Author Response

  1. Abstract and conclusion are identical. I think the authors should rewrite the conclusion. Make it an independent part of the text.

Answer

The abstract and conclusion were rewritten.

  1. There is a typo in the introduction, lines 36, 37 and 38.

Answer

Corrected

  1. The manuscript lacks an experimental part as such. The authors position their article as a research one. In this case, the instrument component and reagents used for research should be indicated. If this is a mini-review, then it must be supplemented with information.

Answer

All experimental procedures, data about materials and instruments, spectra and calculation details, were added in the body of the paper.

  1. In the section describing the reaction mechanism proposed by the authors (Scheme 6), the energy is given in kilocalories per mol. In units of the international system of units, energy should be considered in joules per mole of substance.

Answer Done as recommended

Reviewer 4 Report

It is a nice work and the manuscript was well organized and written. I recommend to accept this manuscript in the present form to be published in Molecules.

The authors present a novel protocol for the synthesis of ethynylpyrroles from decarbonylation of acylethynylpyrroles with acetonitrile in the presence of t-BuOK at room temperature. This protocol showed wide substrate applicability with high yields under mild conditions. The mechanism proposed for the formation of ethynylpyrroles is reasonable. The products were well characterized. However, this protocol suffers from the disadvantages of low atomic economy. Anyway, it's a good manuscript.

Author Response

The reviewer recommentded publication of the paper in the present form

Round 2

Reviewer 1 Report

The authors have addressed the previous comments appropriately and made relevant revisions.